# In Silico and In Vitro Search for Dual Inhibitors of the *Trypanosoma brucei* and *Leishmania major* Pteridine Reductase 1 and Dihydrofolate Reductase

**DOI:** 10.3390/molecules28227526

**Published:** 2023-11-10

**Authors:** Katharina Possart, Fabian C. Herrmann, Joachim Jose, Thomas J. Schmidt

**Affiliations:** 1University of Muenster, Institute for Pharmaceutical Biology and Phytochemistry (IPBP), PharmaCampus, Corrensstrasse 48, D-48149 Muenster, Germany; k_poss01@uni-muenster.de (K.P.); fabian.herrmann@uni-muenster.de (F.C.H.); 2University of Muenster, Institute of Pharmaceutical and Medicinal Chemistry, PharmaCampus, Corrensstrasse 48, D-48149 Muenster, Germany; joachim.jose@uni-muenster.de

**Keywords:** *Trypanosoma brucei*, *Leishmania major*, human African trypanosomiasis, cutaneous leishmaniasis, pteridine reductase 1 inhibitor, dihydrofolate reductase inhibitor, natural products, in silico screening

## Abstract

The parasites *Trypanosoma brucei* (*Tb*) and *Leishmania major* (*Lm*) cause the tropical diseases sleeping sickness, nagana, and cutaneous leishmaniasis. Every year, millions of humans, as well as animals, living in tropical to subtropical climates fall victim to these illnesses’ health threats. The parasites’ frequent drug resistance and widely spread natural reservoirs heavily impede disease prevention and treatment. Due to pteridine auxotrophy, trypanosomatid parasites have developed a peculiar enzyme system consisting of dihydrofolate reductase-thymidylate synthase (DHFR-TS) and pteridine reductase 1 (PTR1) to support cell survival. Extending our previous studies, we conducted a comparative study of the *T*. *brucei* (*Tb*DHFR, *Tb*PTR1) and *L*. *major* (*Lm*DHFR, *Lm*PTR1) enzymes to identify lead structures with a dual inhibitory effect. A pharmacophore-based in silico screening of three natural product databases (approximately 4880 compounds) was performed to preselect possible inhibitors. Building on the in silico results, the inhibitory potential of promising compounds was verified in vitro against the recombinant DHFR and PTR1 of both parasites using spectrophotometric enzyme assays. Twelve compounds were identified as dual inhibitors against the *Tb* enzymes (0.2 μM < IC_50_ < 85.1 μM) and ten against the respective *Lm* enzymes (0.6 μM < IC_50_ < 84.5 μM). These highly promising results may represent the starting point for the future development of new leads and drugs utilizing the trypanosomatid pteridine metabolism as a target.

## 1. Introduction

Approximately 1.9 billion people living in tropical climates are affected by the consequences of neglected tropical diseases (NTDs). Within the diverse array of pathogens responsible for the 20 infectious diseases classified by the World Health Organization (WHO) as NTDs, trypanosomatids form an important subgroup [1]. In continuation of our previous efforts to find natural product inhibitors of crucial target enzymes of trypanosomatid parasites [2,3,4,5], we have focused on the identification of new lead structures with dual inhibitory activity against the corresponding enzymes, namely, pteridine reductase 1 (PTR1) and dihydrofolate reductase (DHFR), of *Trypanosoma brucei* (*T*. *brucei*, *Tb*) and *Leishmania major* (*L*. *major*, *Lm*).

The human pathogenic species *T*. *brucei gambiense* and *T*. *brucei rhodesiense* are the causative agents of human African trypanosomiasis (HAT) or sleeping sickness, which threatens approximately 55 million people in sub-Saharan Africa each year. The parasite is most commonly transmitted during the blood meals of tsetse flies (Glossinidae), at which point it enters, successively, the subcutaneous tissue, the bloodstream, and, after some time, the central nervous system of the host. The course of HAT is fatal without adequate treatment and the currently available drugs often exhibit severe side effects and demand medical supervision during application, which is widely unavailable in the affected regions [6,7,8]. The nitroimidazole derivate fexinidazole is so far the only drug against HAT that allows oral administration [9]. In addition to the human pathogenic subspecies, *T*. *brucei brucei* causes the cattle disease nagana, which poses a major obstacle to the economic development of affected populations living in rural areas that are heavily reliant on well-functioning agriculture and livestock farming [7]. A related species, *Trypanosoma cruzi* (*Tc*), is responsible for Chagas disease, occurring mainly in South and Middle America, and responsible for a considerable burden of disease [7].

Among the approximately 20 human pathogenic *Leishmania* species, *L*. *major* causes the most common form of leishmaniasis, cutaneous leishmaniasis (CL), with at least 200,000 reported cases annually. The genus *Leishmania* comprises intracellular parasites that invade the host’s macrophages after transmission through female sandflies (Phlebotominae). Skin lesions that occur during the course of the infection often result in lifelong scarring, leading to serious handicaps and social stigmatization [10]. CL is most commonly treated by parenteral or intralesional injections of antimonials, which are often painful for the patient and entail a high systemic toxicity [11].

The described challenges in treatment are being further complicated by the high adaptability of both parasites, which enables the formation of broad animal and insect reservoirs, as well as the regular development of drug resistance. The latter, in particular, makes the continuous search for new medication options indispensable [7,8].

As part of evolutionary adaptation, members of the Trypanosomatidae have developed a pteridine auxotrophy, which makes them dependent on the extracellular uptake of pteridines and folates via their hosts to ensure cell survival. As described in detail in our previous works [3,4], the trypanosomatid pteridine metabolism, therefore, represents a highly interesting target for the development of chemotherapeutics against the different forms of trypanosomiasis and leishmaniasis through the inhibition of the oxidoreductase DHFR, as well as its corresponding rescue enzyme, the short-chain dehydrogenase-reductase PTR1. Both investigated enzymes are promising candidates for the design of selective drugs; PTR1, a unique enzyme in the Trypanosomatidae family maintaining the metabolization of pteridines in cases of DHFR inhibition, is completely absent in the human genome. Furthermore, the DHFR enzymes of *T*. *brucei* and *L*. *major* exhibit a relatively low sequence identity of only 26% with the human DHFR (*h*DHFR) [12,13]. Gene knockout experiments have proven that a functioning DHFR and PTR1 are essential in *L*. *major*, requiring a dual inhibition of both enzymes [14]. Similar studies in *T*. *brucei* indicated that the PTR1 itself is vital for parasite survival; thus, its inhibition alone confers vital damage to the parasite. However, the additional inhibition of DHFR can be conceived to exert a more thorough antitrypanosomal effect [15].

Following up on previous results where we identified a selection of sesquiterpene lactones (STLs) as inhibitors of the *T*. *brucei* PTR1 and DHFR [4], a pharmacophore-based in silico screening was conducted with four natural product libraries comprising almost 5000 compounds and a selection of in silico hit compounds was then tested in spectrophotometric enzyme inhibition assays against recombinant *Tb*PTR1/*Tb*DHFR and *Lm*PTR1/*Lm*DHFR. Thereby, we sought to identify further natural products of different biosynthetic classes as dual inhibitors of the respective *T*. *brucei* (*Tb*PTR1/*Tb*DHFR) and *L*. *major* (*Lm*PTR1/*Lm*DHFR) enzymes. Moreover, genus-dependent activity differences in the active compounds towards the respective enzymes were examined in more detail based on the in silico and in vitro results. The selectivity of the most promising inhibitors was further evaluated in vitro in analogous experiments with *h*DHFR.

## 2. Results

### 2.1. Pharmacophore-Based Virtual Screening for Inhibitors of the T. brucei and L. major PTR1 and DHFR

Building on our previous research on *Tb*PTR1 and *Tb*DHFR [2,4], as well as *Lm*PTR1 [3], protein models of *T. brucei* and *L. major* PTR1 and DHFR were selected from the Protein Data Bank (PDB) to perform a pharmacophore-based virtual screening for new inhibitors from natural product databases using the software Molecular Operating Environment (MOE), v. 2018.0101. For this purpose, three structural libraries of commercially available compounds (Phytolab, AnalytiCon Discovery, Specs Natural Products), altogether amounting to approximately 4880 compounds, were used for virtual screening. Based on the resolution of the models (<3 Å) and the respective co-crystallized inhibitors, five protein models were chosen for *Tb*PTR1 and two for *Tb*DHFR. For *Lm*PTR1, four protein structures could be selected. Considering the possible interactions of the co-crystallized inhibitors with the respective binding pockets and the cofactor NADP/NADPH that were calculated by MOE, both complex- and target-based pharmacophores were created for the 11 protein models, amounting to 22 pharmacophore hypotheses. Applying the pharmacophores as filters in the subsequent virtual screening, the resulting hit compounds were further investigated by rigid docking, followed by induced-fit docking simulations and ranked according to their S-score (in kcal/mol). For each examined database, the top ten compounds were identified as hits for the respective enzyme models from which a selection of natural products was tested in vitro (for details, refer to Section 4.1). The compounds that were obtained and tested for each of the 11 protein structures along with the respective calculated S-scores are reported in Appendix A, alongside each of the co-crystallized inhibitors that were employed as templates for the complex-based pharmacophore hypotheses (Appendix A).

To date, there is no protein structure available for *Lm*DHFR in the PDB. The enzyme was therefore not included in the virtual screening. Instead, the 3D structure of the *Lm*DHFR-TS as a bifunctional enzyme was approximated using homology modeling to allow in silico investigations for potential interaction profiles of compounds with in vitro activity against the recombinant protein, as well as comparative studies with the *Tb*DHFR (for details, see Section 4.1.2). The *Lm*DHFR homology model can be found in Appendix A.

### 2.2. In Vitro Evaluation of the In Silico Hits against the T. brucei and L. major PTR1 and DHFR

All target enzymes were prepared by recombinant expression in *Escherichia coli* (*E*. *coli*). The enzyme inhibition assays were established based on previous reports (for details, see Section 4.2) [2,3,4].

Based on the in vitro assays against *Tb*PTR1 and *Lm*PTR1 performed in the previous works of our group, as well as the in silico results obtained from the current study, a total of 47 natural products were tested against the target enzymes. Initially, the relative inhibitory activity of the test compounds was determined at a fixed concentration against the recombinant enzymes (% inhibition of activity at 100 μM for *Tb*PTR1; 50 μM for *Tb*DHFR, *Lm*PTR1, and *Lm*DHFR). In case a compound achieved a relative inhibition of >50%, its half-maximal inhibitory concentration (IC_50_) or half-maximal effective concentration (EC_50_) was determined using concentration-effect curves. The concentration-effect curves for the IC_50_ and EC_50_ values that were obtained in the present study are depicted in Appendix A.

Due to the importance of sufficient PTR1 inhibition, especially in the case of *T. brucei* [15], the in silico hits were first tested against the respective parasites’ PTR1 enzymes. In case of a relative PTR1 inhibition > 50%, the compounds were further tested for their activity against the corresponding DHFR. Furthermore, compounds identified in this way to be inhibitors of *Tb*PTR1 or *Lm*PTR1 were tested against the respective other parasite’s PTR1 in order to elucidate possible genus-dependent affinity differences of the active natural products.

Out of the forty-seven tested natural products, twenty-two compounds achieved >50% inhibitory activity against at least one of the four target enzymes, *Tb*PTR1/*Tb*DHFR or *Lm*PTR1/*Lm*DHFR (hit rate = 46.8%). The molecular structures of the active hits are shown in Figure 1, Figure 2 and Figure 3; the structures of the remaining 25 natural products tested in vitro can be found in Appendix A.

Regarding activity against *Tb*PTR1, 18 natural product inhibitors were identified (compounds **1**, **2**, **5**, **8**–**22**), of which 12 compounds (**2**, **5**, **8**, **9**, **11**, **13**–**17**, **20**, **21**) also inhibited *Tb*DHFR, thus displaying a dual inhibitory effect. For *Lm*PTR1, ten inhibitors were determined (**3**, **4**, **6**–**9**, **11**, **14**, **20**, **22**), of which nine compounds exhibited dual inhibition with activity against *Lm*DHFR (**3**, **4**, **6**–**9**, **11**, **14**, **20**). For 15 compounds, there was enough substance available to be tested in selectivity studies against *h*DHFR. Starting with relative inhibition measurements using a set compound concentration of 50 μM, nine natural products displayed over 50% inhibition (**3**–**6**, **9**, **11**, **14**, **17**, **21**). The obtained IC_50_ and EC_50_ values of each natural product are shown in Table 1.

### 2.3. Investigation of the Mechanism of Inhibition for Selected TbPTR1 Inhibitors

The catalytic center of *Tb*PTR1 contains a cysteine residue (*Tb*Cys168) in close proximity to the substrate/inhibitor binding site. In our previous publication [4], we mentioned the possibility that Sesquiterpene Lactones **1** and **2** might interact as Michael acceptors with *Tb*Cys168 by covalent bond formation with its thiol group [4]. Therefore, we have now subjected a selection of natural product inhibitors containing Michael acceptors (**1**, **2**, **5**, **17**) to initial testing for irreversible inhibition, according to the dilution method described by Bisswanger [16] (see Section 4.2.9).

In this method, the enzyme activity of the respective enzyme-inhibitor mixture is measured before and after a defined dilution (in this study, 1:2). In the absence of an inhibitor, dilution decreases the enzyme activity proportionally to the dilution factor. In the presence of an irreversible inhibitor inactivating part of the enzyme irreversibly (e.g., by covalent binding), the remaining activity is equally reduced by the dilution. A reversible inhibitor, on the other hand, will dissociate upon dilution, resulting in a weaker decrease in activity than what corresponds to the dilution effect. In the presence of Inhibitors **1** and **2**, the decrease in *Tb*PTR1 activity upon the 1:2 dilution was almost proportional to the dilution factor, i.e., only 9.5% and 6.8% activity, respectively, were recovered compared to the reference. This would indeed support an irreversible mechanism of inhibition. In the case of Compounds **5** and **17**, the 1:2 dilution led to somewhat higher activity recoveries of 16.6% and 12.8%; thus, an irreversible inhibition can only be assumed with less certainty. In the case of 5,7-dihydroxy-3,3′,4′,5′,6′,8-hexamethoxyflavone (**23**), which does not have a reactive Michael acceptor structure and was, hence, tested as a very likely positive control for a reversible inhibitor, the dilution experiment indeed led to a recovery of about 33% of the activity. However, further investigations on these compounds’ mechanisms of inhibition are under way.

## 3. Discussion

In the present study, a combination of in silico and in vitro experiments was used to successfully expand our previous works focusing on the enzymes of the trypanosomatid pteridine metabolism by identifying dual natural product inhibitors of *Tb*PTR1/*Tb*DHFR and *Lm*PTR1/*Lm*DHFR [2,3,4].

The *T*. *brucei* and *L*. *major* PTR1 enzymes exhibit rather rigid, sterically restricted catalytic centers. The crystal structures of these enzymes highlight that the formation of a π-sandwich complex between the nicotinamide ring of NADP and a nearby phenylalanine (*Tb*Phe97, *Lm*Phe113) is crucial for the binding of the folate substrate. Various inhibitors compete for this interaction. While the molecular surface of the cofactor binding site is relatively hydrophilic, the substrate/inhibitor binding site ends in an increasingly lipophilic rim. A triad consisting of a serine, a tyrosine, and a lysine (*Tb*Ser95, *Tb*Tyr174, *Tb*Lys178; *Lm*Ser111, *Lm*Tyr194, *Lm*Lys198) located near the cofactor’s nicotinamide moiety plays an important role in the formation of H-bonds with the natural substrate and is well conserved in *Tb*PTR1 and *Lm*PTR1. An important difference between the two parasites’ PTR1 enzymes lies in a cysteine residue (*Tb*Cys168) located near the edge of the substrate binding site of *Tb*PTR1, which could be susceptible to covalent modification by Michael acceptors due to its thiol group. In contrast, *Lm*PTR1 possesses an unreactive leucine residue (*Lm*Leu188) in this position. Compared to PTR1, the *T*. *brucei* and *L*. *major* DHFR binding pockets display a bigger cavity and exhibit higher flexibility. The molecular surface is dominated by lipophilic areas and the natural substrate is almost completely enclosed by the catalytic center. Due to its higher flexibility and size, it can be assumed that the DHFR generally allows more variety in the binding mode and structure of potential inhibitors compared to PTR1. Scaffold-hopping could, therefore, also be more easily achieved for the DHFR than PTR1 [17,18].

We recently reported on the inhibitory activity of the sesquiterpene lactones (STLs) cnicin (**1**) and cynaropicrin (**2**), along with further single and dual enzyme inhibitors of the STL group against the *T*. *brucei* PTR1 and DHFR [4]. Since the in silico docking results indicated that at least one of the Michael acceptor structure elements (i.e., α,β-unsaturated carbonyl structures) of both, Sesquiterpene Lactones **1** and **2**, may be localized near *Tb*Cys168 [4] and since the dilution assay (see Section 2.3) supported an irreversible inhibition mechanism towards *Tb*PTR1 for both compounds, a covalent interaction with the cysteine’s thiol group may underlie these inhibitors’ activity. Based on their promising activity against the target enzymes of *T*. *brucei*, the impact of sesquiterpene lactones was also investigated for the *L*. *major* PTR1 and DHFR. No relevant inhibitory effect of Compounds **1** and **2** could be observed against *Lm*PTR1. In view of the above-mentioned difference between the respective *Tb*PTR1 and *Lm*PTR1 inhibitor binding sites, it is very plausible that the lack of reactivity of *Lm*Leu188 compared to *Tb*Cys168 towards the Michael acceptors is responsible for the observed inactivity of these compounds against the *Leishmania* enzyme. Further experimental investigation of this hypothesis and the possible exploitation of Cys168 in *Tb*PTR1 as a drug target are the subject of ongoing studies, including investigations on inhibition kinetics, mass spectral measurements, and site-directed mutagenesis experiments. It is noteworthy, that no relevant inhibition of *h*DHFR was observed for STLs **1** and **2**; thus, they can be considered selective inhibitors of the *T*. *brucei* PTR1/DHFR.

In addition to STLs, the diarylheptanoid dehydrohirsutanone (**17**, for docking conformation, see Figure 4) and the structurally related curcumin (**5**) were identified as dual inhibitors of the *Tb* enzymes, with Compound **17** achieving the lowest IC_50_ against *Tb*PTR1 at 8.3 μM. Both compounds had also displayed in vitro activity against *T*. *brucei* in the past (EC_50_ = 2.5 μM, for both compounds, respectively) [19]. Compound **5** also displayed moderate activity against *L*. *major* (EC_50_ = 33.0 μM); whereas, Compound **17** showed weak to no inhibition (EC_50_ > 100 μM) [19].

Compounds **5** and **17** were also tested against *Lm*PTR1 and showed significantly less activity than against the *Tb* enzyme, which would support the importance of their Michael acceptor centers in interacting with the above-mentioned cysteine of *Tb*PTR1 (compare Figure 4). Since, however, the irreversible inhibition of *Tb*PTR1 could not be so clearly supported for Compounds **5** and **17** via the dilution method (see Section 2.3), additional studies of their inhibition mechanism have been initiated. Furthermore, the rather unfavorable pharmacokinetic and pharmacodynamic properties of curcumin analogs would probably require further optimization of the molecular structure [20]. In addition, curcumin and dehydrohirsutanone showed some activity against human DHFR (IC_50_ = 33.9 and 40.7 μM, respectively); thus, they would also require some optimization with respect to selectivity.

The anacardic acids Compounds **11** and **12** were initially identified as *Tb*PTR1 inhibitors in silico, forming a π-sandwich complex between their salicylic acid substructure and the pyridine of NADP as well as the adjacent phenylalanine during docking simulations. In the binding mode postulated by docking in MOE, their phenolic OH group provides a hydrogen bond donor for the adjacent oxygen of the phosphate linker of NADP. The predominantly saturated hydrocarbon chain of both compounds is orientated in the lipophilic region at the edge of the binding pocket, suggesting hydrophobic interactions (see Figure 5).

IC_50_ values of 20.1 μM and 21.7 μM were achieved with Compounds **11** and **12** against *Tb*PTR1, respectively. Compound **11** was also identified as a *Tb*DHFR inhibitor (IC_50_ = 0.2 μM), as well as a dual inhibitor against the *Lm*PTR1 (IC_50_ = 10.2 μM) and *Lm*DHFR (IC_50_ = 2.6 μM). However, its particularly strong inhibition of the parasites’ DHFR enzymes does not appear particularly selective since it also inhibits the human enzyme, with a low IC_50_ of 2.4 μM. Compound **12**, possessing a structure almost identical to Compound **11**, unfortunately, could not be tested against *Tb*DHFR and against the *Lm* enzymes due to the very limited sample amount available. In past *Tb*PTR1 studies, several anacardic acid derivatives displayed activity against the glyceraldehyde-3-phosphate dehydrogenase (GAPDH) of *T*. *cruzi* (IC_50_ = 25–55 μM) [21].

Further, (poly)phenolic secondary metabolites (**3**, **4**, **7**–**10**, **16**, **20**–**22**) could be identified as inhibitors against the enzymes under study. Dual inhibitory properties of flavonols, a subgroup of polyphenols, against the *Tb*PTR1/*Tb*DHFR, have already been described in the past [3,22]. According to the protein-inhibitor interactions that were postulated in silico, it can be assumed that the inhibitory potential of polyphenols is partly rooted in their ability to form the aforementioned π-sandwich complex in the PTR1 binding pocket. Among this compound group, the stilbene derivative salvianolic acid A (**8**), the flavanone sophoraflavanone G (**9**), a chalcone derivative (**10**), surangin B (**20**), and acrovestone (**21**) exhibited dual inhibition against *Tb*PTR1 and *Tb*DHFR. Antitrypanosomal activity in *T*. *brucei* cell assays could already be observed for Compounds **8** (IC_50_ = 3.1 μM) and **9** (IC_50_ = 1.4 μM) in previous studies [2,23]. Against the *L*. *major* enzymes (−)-catchin-3-gallate (**3**), the flavanolignan 2,3-dehydrosilybin B (**4**), as well as the chalcone derivative isoliquiritin (**7**), salvianolic acid A (**8**), sophoraflavanone G (**9**), and surangin B (**20**) showed dual inhibition, with Compound **9** achieving the best activity against *Lm*PTR1 (IC_50_ = 19.2 μM) [3] and Compound **3** exhibiting the highest inhibition against *Lm*DHFR (IC_50_ = 0.6 μM). However, some of these active phenolics, Compounds **3**, **4**, **9**, and **21**, also displayed significant activity against *h*DHFR; therefore, these inhibitors may suffer from selectivity issues. In contrast, selective inhibition of the parasite enzymes was observed for the Phenolic Compounds **7**, **8**, **20**, and **22**.

Finally, a variety of prenylated xanthone derivatives showed potential for dual inhibition. Several xanthones have already exhibited inhibitory activity against *Plasmodium falciparum* and *Leishmania mexicana* [24,25]. In the present study, Compounds **13**–**15** inhibited both *Tb* enzymes with IC_50_ values ranging between 71.0 and 74.4 μM against *Tb*PTR1 and between 0.2 and 2.9 μM against *Tb*DHFR. Against the respective *Lm* enzymes, both garcinone c (**6**) and g-mangostin (**14**) displayed promising activity, with Compound **14** being the more potent inhibitor with an IC_50_ of 10.0 μM against *Lm*PTR1 and 1.5 μM against *Lm*DHFR. However, Compounds **6** and **14** also displayed relatively low IC_50_ values of 1.7 and 2.7 μM, respectively, against *h*DHFR, which indicates that structural modifications to these substances would be required to improve their selectivity.

## 4. Materials and Methods

The experimental section of this publication is building on our previous works on this project [2,3,4]. Unless stated otherwise, please refer to [4] for a detailed description of the overlapping in silico and in vitro protocols.

### 4.1. In Silico Procedure

All in silico studies were performed with the software Molecular Operating Environment v. 2018.0101 (MOE, Chemical Computing Group, Montreal, QC, Canada) under the conditions of the implemented force field MMFF94x (Merck Molecular Force Field) [26].

#### 4.1.1. Preparation of the Respective 3D Protein Structures

Searching for 3D models of the enzymes PTR1 and DHFR from the family Trypanosomatidae available in the Protein Data Bank (PDB) of the Research Collaboratory for Structural Bioinformatics (RCSB) [27], five suitable structures with highly active inhibitors as ligands could be selected in case of *Tb*PTR1 (PDB entries “2X9G”, “3MCV”, “4CMJ”, “4CMK”, “5JDI” [28,29,30]) and two for *Tb*DHFR (“3QFX”, “3RG9” [31,32]). For the in silico studies on *L*. *major*, four structures were chosen for *Lm*PTR1 (“1E7W”, “1W0C”, “2BFM” and “2QHX” [17,33,34,35]). Lacking an experimentally determined protein structure for *Lm*DHFR, a homology model was calculated with MOE based on the 3D-structure of the *Trypanosoma cruzi* (*T*. *cruzi*, *Tc*) DHFR-TS model “3KJS” [36]. Following the procedure described in previous works [4], the chosen protein models were structurally corrected and the energy was minimized before their application in the following in silico experiments.

#### 4.1.2. Homology Modeling

Due to the unknown 3D structure of the *L*. *major* DHFR-TS, a homology model of the protein was generated using MOE. The target amino acid sequence of *Lm*DHFR-TS was taken from the GenBank of the National Centre for Biotechnology Information (NCBI) (*L. major* DHFR-TS, Gene ID: 5649109). The crystal structure “3KJS” of *T. cruzi* DHFR-TS was chosen using a template from a selection of homologous protein models from the PDB (MOE: Protein → Search → PDB) and optimized according to Section 4.1.1. This selection was based on the evolutionary relationship of *Lm* and *Tc* and the high sequence similarity and identity of 76.8% and 66.3%, respectively (see Appendix A). The amino acid sequences of the target and template were aligned pairwise (MOE: Sequence Editor → Alignment → Align/Superpose). Lastly, the homology model for *Lm*DHFR-TS was created based on the template structure and processed according to Section 4.1.1 (MOE: Sequence Editor → Protein → Homology Model). The originally co-crystallized cofactor NADPH and the ligand from “3KJS” were transferred to the homology model to prevent a collapse in the binding pocket during the computational optimization and energy minimization of the model.

For evaluation, the 3D structures of the *Tc*DHFR-TS model and the generated *Lm*DHFR-TS homology model were superposed. Their sequence identity, sequence similarity, and root-mean-square deviation (RMSD, ≤2.0 Å) were calculated. A reasonable geometry of the homology model was further ensured by examining its Ramachandran plot (MOE: Sequence Editor → Protein → Geometry → Phi-Psi Plot) in which no major deviations from the common geometric preferences of amino acids in proteins were observed.

#### 4.1.3. Pharmacophore Design

Two different pharmacophore hypotheses were generated based on each of the eleven *Tb*PTR1, *Tb*DHFR, and *Lm*PTR1 protein models [4]. The nature and localization of the interactions of the respective co-crystallized ligands with the binding pockets were investigated to create complex-based pharmacophores. For target-based pharmacophores, the ligands were masked out and inferences about potential interactions were drawn from the structural makeup of the receptor’s amino acids alone. The potential interactions with the coenzyme NADP/NADPH, which was co-crystallized in every model, were considered in both approaches.

#### 4.1.4. Virtual Screening of Natural Product Databases

Focusing on the identification of natural products, as well as their derivatives with inhibitory activity against the target enzymes, we selected the natural product databases of Phytolab GmbH (Vestenbergsgreuth, Germany; 1500 compounds), AnalytiCon Discovery GmbH (Potsdam, Germany; 5000 compounds), and Specs Natural Products (Zoetermeer, Netherlands; 744 compounds) for the virtual screening (VS). All database compounds were prepared for VS in MOE prior to their use and filtered for drug-like natural products, according to Lipinski’s “Rule of five” [37], overall, amounting to ca. 4880 compounds. The VS was carried out as described in previous works [4], using the pharmacophore hypotheses generated in Section 4.1.3 as filters.

#### 4.1.5. Molecular Docking

The natural product hits identified via VS were subsequently examined and validated in more detail during a two-stage docking simulation to predict the preferred orientation of the interacting molecular species [4]. Initially, all compounds underwent a docking process with a rigid receptor, with the small-molecule natural products being positioned in different conformations and tautomers in the respective catalytic center. Based on the results of the rigid docking, the ten best compounds for each pharmacophore model were selected, based on their S-score, to perform a docking process in induced-fit mode, simulating the interactions of the compounds with a flexible binding pocket. From this process, five top hits were determined for each of the fourteen target- and complex-based pharmacophore models for *Tb*PTR1 and *Tb*DHFR and the eight pharmacophores for *Lm*PTR1 from the individual natural product databases. The resulting compounds were then considered for in vitro investigation in a spectrophotometric enzyme inhibition assay and a selection was made according to the diversity of the molecular structures and substance availability.

### 4.2. In Vitro Procedure

#### 4.2.1. Cloning of *Tb*DHFR and *Lm*DHFR into the *E. coli* BL21(DE3) Host Strain

For a detailed description of the pET 11D::*Tb*DHFRHis vector design for the recombinant overexpression of *Tb*DHFR in *E*. *coli*, please refer to [4].

The preparation of the vector pET 11D::*Lm*DHFRHis for the expression of *Lm*DHFR was performed analogously. The required gene sequence of *L. major* DHFR-TS (Gene ID: 5649109) was taken from the NCBI GenBank server. The DNA fragment encoding *Lm*DHFR was amplified by Phusion DNA polymerase (Thermo Fisher Scientific, Bonn, Germany) by using the forward and reverse primers KP08 (5′-CACCATCACCATCATATGAGCCGTGCAGCAGC-3′) and KP09 (5′-CAGCCGGATCCGTTAATTGCGAGGCACATATTTACAG-3′) (Eurofins MGW Operon, Ebersberg, Germany). The plasmid backbone pET 11D-kduD (Merck, Darmstadt, Germany) [38] was amplified with the primers SB001 (5′-TAACGGATCCGGCTGCTAAC-3′) and MS41 (5′-ATGATGGTGATGGTGGTGCATG-3′) and the original template DNA removed through DpnI digestion. Using a 1% agarose gel (110 V for 50 min), the PCR products were separated and subsequently purified with the QIAquick Gel Extraction Kit (Qiagen, Hilden, Germany). The desired plasmid was assembled via In-Fusion cloning (In-Fusion HD EcoDry, Clontech, Saint-Germain-en-Laye, France) and transformed into competent *E. coli* Stellar cells (Invitrogen). Positive clones were verified via agarose gel electrophoresis and the plasmid was isolated using the innuPREP Plasmid Mini Kit (Analytik Jena, Jena, Germany). The pET 11D::*Lm*DHFRHis construct was analyzed by Seqlab (Goettingen, Germany) and encodes for *Lm*DHFR controlled by a T7/lac promoter with a N terminal His6 tag and a carbenicillin resistance gene.

#### 4.2.2. Recombinant Expression and Purification of *Lm*PTR1 and *Tb*PTR1

The heterologous expression of the *L. major* and *T. brucei* PTR1 was carried out using already transformed *E*. *coli* BL21 (DE3) strains provided by the working group of Prof. Dr. M. Paola Costi (Modena, Italy). The strains contained the vectors pET 15b::*Lm*PTR1His and pET 15b::*Tb*PTR1His, respectively, which encode for either *Lm*PTR1 or *Tb*PTR1 controlled by the T7/lac promotor and include a N-terminal His6 tag and a carbenicillin resistance gene.

The recombinant *Lm*PTR1 and *Tb*PTR1 were cultivated and purified according to our previous works, following a modified procedure by Sambrook and Russell [39]. The culture was used to inoculate 1 L Erlenmeyer flasks with 200 mL of LB medium (1:1000) and incubated until an optical density (OD_578nm_) of 0.6 to 0.9 was reached (5 h, 37 °C, 200 rpm). Carbenicillin was added to all cultures at a concentration of 50 μg/mL. Expression was induced by adding 0.4 mM of isopropyl-*β*-D-thiogalactopyranoside (IPTG) to the culture, followed by incubation (16 h, 18 °C, 200 rpm). After the induction of the cell culture was completed, the *E*. *coli* cells were harvested and resuspended in lysis buffer. Following cell disruption, the soluble fraction of the lysate was retrieved via centrifugation. Immobilized Metal Ion Affinity Chromatography (IMAC) was applied as a purification method to separate the respective enzymes from the crude extract, using a nitrilotriacetate (NTA-Ni^2+^) loaded column. The fractions containing target proteins were identified by polyacrylamide gel electrophoresis (SDS PAGE, 12.5%) and dialyzed for 4 h at 4 °C (50 mM Tris/HCl (pH 7.6), 100 mM NaCl). The resulting lysate containing *Tb*PTR1 or *Lm*PTR1, respectively, was mixed with 20% glycerol for cryoprotection and stored in aliquots at −80 °C.

#### 4.2.3. Recombinant Expression and Purification of *Lm*DHFR and *Tb*DHFR

The expression and purification procedures for *Lm*DHFR and *Tb*DHFR were performed analogously to Section 4.2.2. To create a reducing environment, 2-mercaptoethanol (BME, 7 mM) was added to the lysis buffer during the resuspension process of the cell pellets. The fusion proteins were separated and purified from the crude extract via IMAC. A 12.5% SDS-PAGE was employed to monitor fractions containing the target protein, which were pooled and then dialyzed in reducing conditions for 4 h at 4 °C (50 mM Tris/HCl (pH 7.6), 100 mM NaCl, 10 mM dithiothreitol (DTT)). The purified enzymes were supplemented with 20% glycerol and stored in aliquots at −80 °C.

#### 4.2.4. Recombinant Expression and Purification of *h*DHFR

The pET 15b::*h*DHFRHis vector for the recombinant expression of *h*DHFR in *E*. *coli* was obtained from BioCat GmbH (Heidelberg, Germany) and 15b::*h*DHFRHis was transformed into an *E*. *coli* BL21 (DE3) strain and cultivated overnight. The culture was used to inoculate 1 L Erlenmeyer flasks with 200 mL of LB medium (1:1000) and incubated until an optical density (OD_578nm_) of 0.6 to 0.9 was reached (5 h, 37 °C, 200 rpm). All cultures were supplemented with 50 μg/mL of carbenicillin. Expression was induced by adding 0.4 mM of IPTG to the culture, followed by incubation (6 h, 30 °C, 200 rpm). The harvest and purification processes for *h*DHFR were performed analogously to Section 4.2.3.

#### 4.2.5. Kinetic Characterization

The respective protein concentration and activity of the expressed enzymes, as well as the saturating conditions of their substrates and co-substrates, were determined by monitoring the oxidation of NADPH to NADP^+^ at 340 nm using UV/Vis spectroscopy (Hitachi U-2900, Tokyo, Japan). The measurements were carried out as triplicates over a time span of 250 s at 30 °C.

##### *Tb*PTR1 and *Lm*PTR1

The used *Tb*PTR1 concentration was 3.23 mg/mL, with a specific activity of 0.03 U/mg. The measurements were carried out using the enzyme’s individual saturating concentrations for folic acid (8 μM) and NADPH (150 μM) in Buffer A (50 mM Tris/HCl (pH 7.6), 250 mM NaCl). For *Lm*PTR1, the protein concentration was 5.02 mg/mL and the specific activity was calculated to be 0.29 U/mg. Buffer B (50 mM NaH_2_PO_4_ (pH 6.0), 100 mM NaCl) was used for all *Lm*PTR1 measurements and the saturating concentrations amounted to 50 μM of folic acid and 200 μM of NADPH. The saturating conditions of the PTR1 enzymes were determined using the diagrams depicted in Appendix A.

##### *Tb*DHFR and *Lm*DHFR

The *Tb*DHFR concentration was measured to be 0.04 mg/mL and a specific activity of 38.1 U/mg was calculated. The concentration of *Lm*DHFR was 0.05 mg/mL with a specific activity of 56.2 U/mg. All measurements were carried out using the saturating concentrations for dihydrofolate (50 μM) and NADPH (150 μM) in Buffer C (50 mM Tris/HCl (pH 7.6), 250 mM NaCl, 10 mM BME). The saturating conditions for *Tb*DHFR and *Lm*DHFR were determined using the diagrams depicted in Appendix A.

##### *h*DHFR

The used *h*DHFR concentration was 0.64 mg/mL, with a specific activity of 0.62 U/mg. All measurements were carried out using the saturating concentrations for dihydrofolate (100 μM) and NADPH (50 μM) in Buffer C (50 mM Tris/HCl (pH 7.6), 250 mM NaCl, 10 mM BME). The saturating conditions for *h*DHFR were determined using the diagrams depicted in Appendix A.

#### 4.2.6. Test Compounds

Compounds **1**–**10**, **14**, and **23**–**31** from Phytolab GmbH (Vestenbergsgreuth, Germany) were kindly donated by the company to support our research and conform to a purity of ≥95%. Compounds **11**–**13**, **15**–**17**, and **32**–**40** were purchased from AnalytiCon Discovery; Compounds **18**–**22** and **41**–**47** were obtained from Specs. The purity of the AnalytiCon Discovery and Specs NP compounds was 90% and ≥95%, respectively, according to the manufacturers‘ specifications.

#### 4.2.7. Single-Concentration Enzyme Inhibition Assays

The inhibitory effect of the selected natural products was investigated by testing the DMSO solution of each compound at a constant concentration of 100 μM for *Tb*PTR1 and 50 μM for *Lm*PTR1, *Tb*DHFR, and *Lm*DHFR, as well as *h*DHFR. The concentrations of the cofactor and substrates were set at the previously determined saturating conditions (Section 4.2.5). Each test compound was incubated for 20 min with the respective enzyme and NADPH to ensure a sufficient interaction time before the reaction was induced by the addition of the substrate. The enzymatic activity was determined in duplicates and correlated with a reference that contained just DMSO, instead of an inhibitor. The enzymatic activity of the reference was normalized to 100%. In the case of a relative inhibition > 50%, the inhibitory effect of the test substances was further characterized by determining IC_50_ values.

#### 4.2.8. Determination of the IC_50_ Values

The half-maximal inhibitory concentration (IC_50_) of the natural products was determined in triplicate, using at least five different inhibitor concentrations against a reference containing no inhibitor. The enzymatic activity was documented and analyzed and the IC_50_ value was determined by nonlinear regression using GraphPad Prism 9 (GraphPad Software Inc., La Jolla, CA, USA). If an enzyme inhibition of 100% could not be achieved experimentally, the half-maximal effective concentration (EC_50_) was determined instead.

#### 4.2.9. Dilution Assay

To assign an either reversible or irreversible inhibition mechanism to selected compounds, the enzymatic activity of an enzyme-inhibitor mixture was determined before and after a defined dilution, using the reaction conditions described in Section 4.2.5 [16].

Two equivalent solutions with the respective enzyme and inhibitor combination were prepared, one of them being diluted with buffer in a 1:2 ratio shortly before the reaction was started by adding the substrate. The respective inhibitor concentration was set near the IC_50_ value. The substrate and cofactor concentrations were kept constant in both reaction mixtures. All measurements were carried out in duplicates against a reference containing DMSO instead of the inhibitor. In case a compound inhibited the enzyme through irreversible binding, the dilution showed proportionally reduced enzyme activity compared to the likewise diluted DMSO reference. If a reversible inhibitor was present, a higher degree of activity recovery (>>10%) could be observed than what corresponds to the dilution factor due to the partial dissociation of the inhibitor from the enzyme upon dilution. The percentual changes in the enzymatic activity of the undiluted and diluted test solution were compared and evaluated to decide the mechanism of inhibition.

## 5. Conclusions

This study allowed the rational selection of 47 test substances from an in silico screening set of 4880 natural products, of which 21 substances showed activity against the target enzymes. Twelve dual inhibitors of *Tb*PTR1/*Tb*DHFR and six single enzyme inhibitors of *Tb*PTR1 were identified. Likewise, nine dual inhibitors against *Lm*PTR1/*Lm*DHFR and one single enzyme inhibitor against *Lm*PTR1 were identified. Experiments on recombinant *h*DHFR further allowed an assessment of the active inhibitors’ selectivity towards the parasite enzymes. Overall, some of the inhibitors identified in this study represent promising starting points for the further development of more active and selective drug leads targeting the trypanosomatid parasites’ pteridine metabolism. Since the related trypanosomatid parasite *T*. *cruzi* also exhibits pteridine auxotrophy and its corresponding enzymes, *Tc*DHFR-TS, *Tc*PTR1, and *Tc*PTR2, have been proven essential for its pathogenicity and survival, it would be an interesting further subject for related future studies [35,36,40].

## Figures and Tables

**Figure 1 molecules-28-07526-f001:**
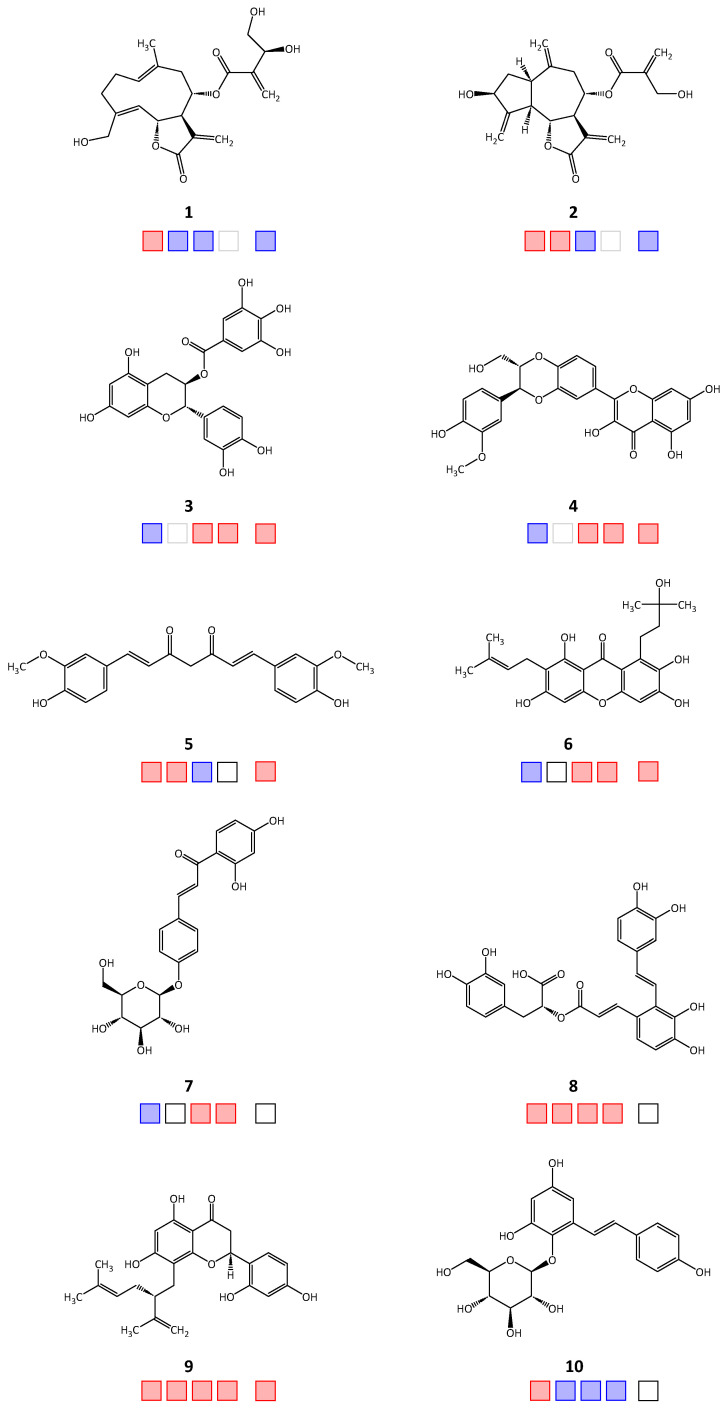
Chemical structures **1**–**10** of the in silico hits identified from the Phytolab database that displayed in vitro inhibitory activity against the target enzymes. Squares from left to right: *Tb*PTR1, *Tb*DHFR, *Lm*PTR1, *Lm*DHFR, *h*DHFR; red: active; blue: inactive; empty: not tested.

**Figure 2 molecules-28-07526-f002:**
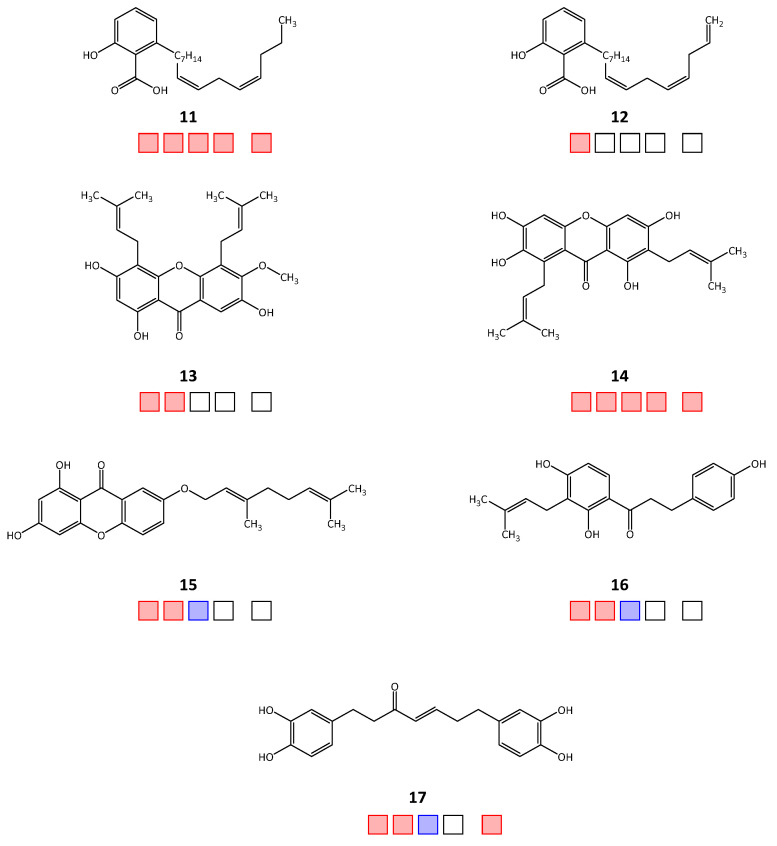
Chemical structures **11**–**17** of the in silico hits identified from the AnalytiCon Discovery database that displayed in vitro inhibitory activity against the target enzymes. Squares from left to right: *Tb*PTR1, *Tb*DHFR, *Lm*PTR1, *Lm*DHFR, *h*DHFR; red: active; blue: inactive; empty: not tested.

**Figure 3 molecules-28-07526-f003:**
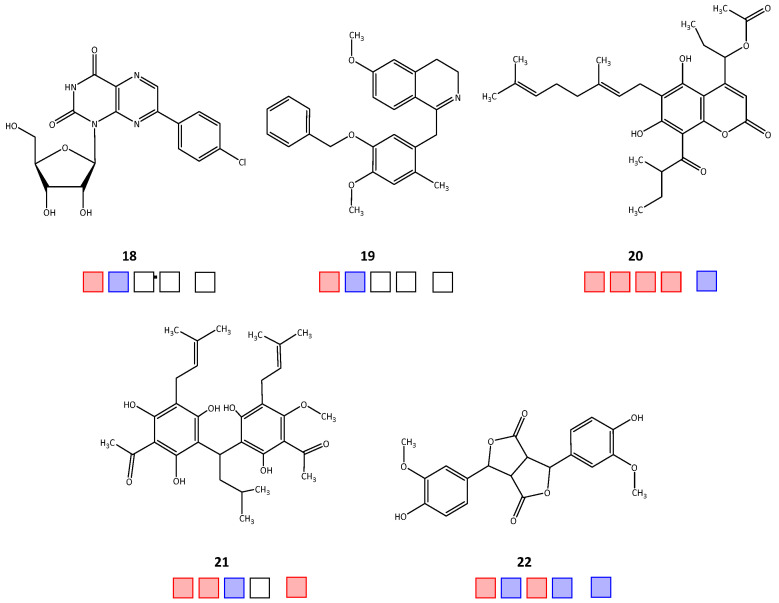
Chemical structures **18**–**22** of the in silico hits identified from the Specs Natural Products (Specs NP) database that displayed in vitro inhibitory activity against the target enzymes. Squares from left to right: *Tb*PTR1, *Tb*DHFR, *Lm*PTR1, *Lm*DHFR, *h*DHFR; red: active; blue: inactive; empty: not tested.

**Figure 4 molecules-28-07526-f004:**
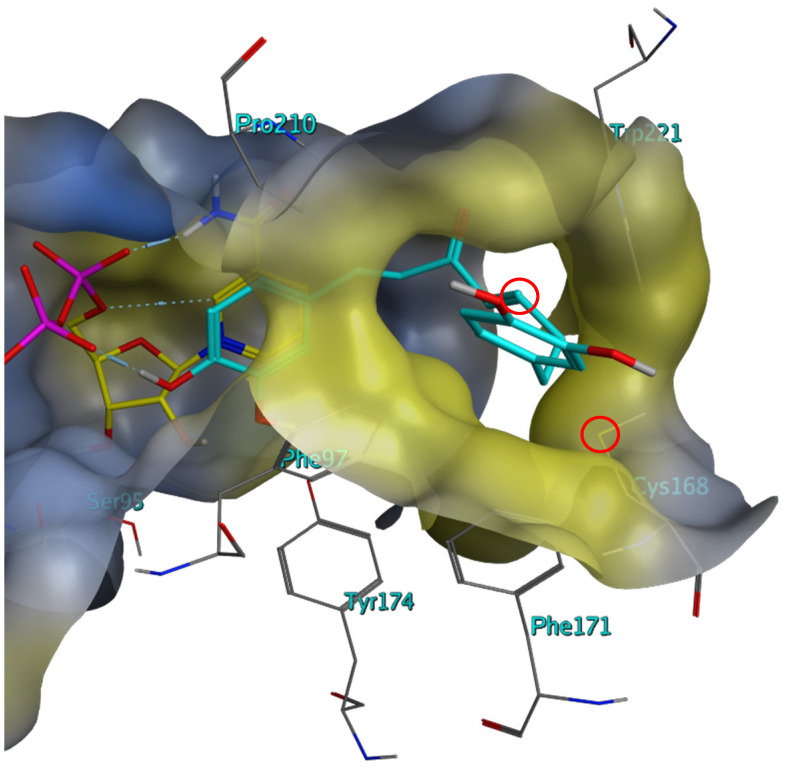
Best scoring docking conformation for Compound **17** in the binding pocket of *Tb*PTR1 (ID: “2X9G”) with co-crystallized NADP (carbon atoms colored in yellow). The molecular surface is colored according to lipophilicity, with lipophilic areas in yellow and hydrophilic areas in blue. Co-crystallized solvent not shown. Note that the reactive β-carbon of the enone system is not far away from the SH group of Cys168 in this docking pose (red circles; 4.6 Å). From this orientation, it could easily assume a position suitable for a Michael addition without much change.

**Figure 5 molecules-28-07526-f005:**
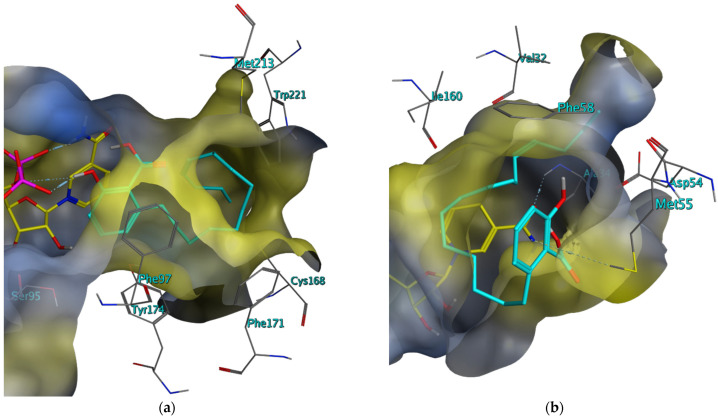
Best scoring docking conformations for Compound **11** in the binding pocket of (**a**) *Tb*PTR1 (ID: “4CMK”) and (**b**) *Tb*DHFR (ID: “3QFX”) with co-crystallized NADP/NADPH (carbon atoms colored in yellow). The molecular surface is colored according to lipophilicity, with lipophilic areas in yellow and hydrophilic areas in blue. Co-crystallized solvent not shown.

**Table 1 molecules-28-07526-t001:** Inhibitory activity of the natural products tested in vitro against *Tb*PTR1/*Tb*DHFR and *Lm*PTR1/*Lm*DHFR. The IC_50_/EC_50_ values that were obtained from previous studies by our group are quoted accordingly and listed for completeness. The IC_50_/EC_50_ values that were determined for *h*DHFR are listed for comparison. Empty fields indicate that the measurements were not continued due to the low inhibitory activity of the compound against parasitic PTR1 (>100 and >50 μM, respectively).

Compound	IC_50_ [μM]				
	*Tb*PTR1	*Tb*DHFR	*Lm*PTR1	*Lm*DHFR	*h*DHFR
**1**	12.2 ^a^ [4]	>50	>50		>50
**2**	12.4 [4]	7.12	>50		>50
**3**	>100 [2]		75.3 [3]	0.6	4.2
**4**	>100		42.9 [3]	2.6	3.8 ^a^
**5**	18.5	10.1	>50		33.9
**6**	>100 [2]		26.3 [3]	3.0	1.7 ^a^
**7**	>100		84.5 [3]	46.2	>50
**8**	85.1	2.5	42.2 [3]	18.7 ^a^	>50
**9**	31.8	9.4	19.2 [3]	14	31.6 ^a^
**10**	83.6	>50	>50	>50	n.t.
**11**	20.1	0.2	10.2	2.6	2.4
**12**	21.7	n.t.	n.t.	n.t.	n.t.
**13**	74.4	0.6	n.t.	n.t.	n.t.
**14**	71.0	0.2	10.0	1.5	2.7
**15**	32.4 ^a^	2.9	>50		n.t.
**16**	64.6	11.7	>50		n.t.
**17**	8.3	17.7	>50		40.7
**18**	58.6	>50	n.t.	n.t.	n.t.
**19**	75.1	>50	n.t.	n.t.	n.t.
**20**	28.3	2.9	35.1 ^a^	4.7 ^a^	>50
**21**	36.8 ^a^	2.1	>50		19.8
**22**	37.8 ^a^	>50	30.2	>50	>50

^a^ EC_50_ values; n.t.: not tested due to small sample amount.

## Data Availability

All molecular modeling data, as well as raw data of the enzyme inhibition study, are available from the corresponding author on request.

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
