# Peer review of "In Silico and In Vitro Search for Dual Inhibitors of the Trypanosoma brucei and Leishmania major Pteridine Reductase 1 and Dihydrofolate Reductase"

_molecules, 2023, doi:10.3390/molecules28227526_

Round 1
Reviewer 1 Report
Comments and Suggestions for Authors
This manuscript by Possart K and colleagues reports their study on inhibitors of pteridine reductase 1 and dihydrofolate reductase of Trypanosoma brucei and Leishmania major using a combination of in silico and in vitro methods. They screened almost 5,000 natural products from three different providers, and identified 21 compounds that show activities against the two targeted enzymes by in silico analyses. At the end after in vitro tests they had found 12 and nine dual inhibitors for the two enzymes of T. brucei and L major. In addition, they found six and one products only against pteridine reductase 1, respectively, of the parasites. The manuscript is well fit to the journal and is worthy of publication.
One major concern is that the authors have never mentioned the 3rd parasite belonging to the group, collectively called “tritryps”, which is T. cruzi. The latter cause Chagas disease in Soth America, and its effect to human health is much greater than T. brucei. The author should at least mention the parasite in introduction, and have discussed what the identified compounds may have on the its corresponding enzymes.
Minor issues:
Family names such as Glossinidae (Line 36) and Phlebotominae (Line 60) shouldn’t be italicized. Only names of genus and species are.
L101: Trypanosoma brucei and Leishmania major should be shortened as T. brucei and L. major. Please make same changes in the rest of manuscript.
The paragraphs ended in Line 309 and starting in Line 310 should be one paragraph.
Delete Lines 196-7 and the empty page.
Author Response
Reviewer 1
This manuscript by Possart K and colleagues reports their study on inhibitors of pteridine reductase 1 and dihydrofolate reductase of Trypanosoma brucei and Leishmania major using a combination of in silico and in vitro methods. They screened almost 5,000 natural products from three different providers, and identified 21 compounds that show activities against the two targeted enzymes by in silico analyses. At the end after in vitro tests they had found 12 and nine dual inhibitors for the two enzymes of T. brucei and L major. In addition, they found six and one products only against pteridine reductase 1, respectively, of the parasites. The manuscript is well fit to the journal and is worthy of publication.
One major concern is that the authors have never mentioned the 3rd parasite belonging to the group, collectively called “tritryps”, which is T. cruzi. The latter cause Chagas disease in Soth America, and its effect to human health is much greater than T. brucei. The author should at least mention the parasite in introduction, and have discussed what the identified compounds may have on the its corresponding enzymes.
A sentence on T. cruzi and its considerable disease burden was added in the introduction. A short statement on this parasite was also added at the end of the conclusions. We hope the reviewer is satisfied with these changes.
Minor issues:
Family names such as Glossinidae (Line 36) and Phlebotominae (Line 60) shouldn’t be italicized. Only names of genus and species are.
This was corrected.
L101: Trypanosoma brucei and Leishmania major should be shortened as T. brucei and L. major. Please make same changes in the rest of manuscript.
This was done.
The paragraphs ended in Line 309 and starting in Line 310 should be one paragraph.
This was corrected.
Delete Lines 196-7 and the empty page.
This was only a problem with the pdf file.
We thank the reviewer for the time and effort to help us improve our manuscript.

Reviewer 2 Report
Comments and Suggestions for Authors
Present work explores the rational search for dual inhibitors of T. brucei and L. major enzimes. The experiments design are adecquates and the development of the studies and results presentation and discussion is coherent, helping readers to understand all the process. The manuscript is suitable for publication in Molecules after minor changes to improve the quality of this good work:
Abstract must be condensed to only one paragraph.
Introductory paragraph from lines 37-42 must be deleted from the beginning of the introduction section and combined with the last paragraph of the section.
Please, extend the number of cites, especially in the introduction section. There are only 39 cites and a several of them are auto cites.
Duplication of sentences in line 180, with “Table 1” missing in the first one.
I recommend accept after minor revision.
Author Response
Reviewer 2
Present work explores the rational search for dual inhibitors of T. brucei and L. major enzimes. The experiments design are adecquates and the development of the studies and results presentation and discussion is coherent, helping readers to understand all the process. The manuscript is suitable for publication in Molecules after minor changes to improve the quality of this good work:
Abstract must be condensed to only one paragraph.
The abstract is now a single paragraph.
Introductory paragraph from lines 37-42 must be deleted from the beginning of the introduction section and combined with the last paragraph of the section.
We do not agree with this suggested change. It is merely a matter of taste where to put this sentence.
Please, extend the number of cites, especially in the introduction section. There are only 39 cites and a several of them are auto cites.
We do not agree with this suggestion since it is not necessary to cite more work and would just mean to inflate the given introduction. Again, this is a matter of taste. Obviously, the number of self-citations passed the rigorous check by the editorial office.
Duplication of sentences in line 180, with “Table 1” missing in the first one.
We agree that the second sentence is not necessary. It was deleted.
I recommend accept after minor revision.
We thank the reviewer for the time and effort to help us improve our manuscript

Reviewer 3 Report
Comments and Suggestions for Authors
The paper by Possart et al. describes the identification of molecules inhibiting enzymes of Trypanosoma brucei and Leishmania major pteridine metabolism, specifically pteridine reductase (PTR1) and dihydrofolate reductase (DHFR). These enzymes are essential to the parasites and represent interesting therapeutic targets to treat sleeping sickness, cutaneous leishmaniasis, and other diseases caused by these trypanosomatid parasites. This is particularly important since the few currently available drugs exhibit severe side effects.
The study expands a previous work of a research group with extensive experience in in silico screening and in vitro selection of inhibitors. I enjoyed reading the manuscript. It is clear, well-written, elaborates well on the obtained data, and discusses them. The conclusions are well supported by the data, and the control experiments are appropriate.
The results are interesting, and the information provided can be useful for numerous researchers, as diseases caused by trypanosomatid parasites remain a serious health problem, and new, more specific drugs are urgently needed. Therefore, I recommend the publication of the manuscript after revision.
Points to improve before publication:
I would suggest including a more detailed description of the experiments needed to determine the type of inhibition of the identified products in the discussion section. While analyzing the effect of dilution of the enzyme preparation on the activity is useful, I would recommend describing other well-known methods (for instance, determining the kinetic parameters in the presence and absence of inhibitors. Vmax would decrease in the presence of an irreversible inhibitor, but Kms would remain unchanged. In this way, irreversible inhibitors behave as reversible non-competitive inhibitors. Other methods like gel filtration, dialysis, etc., could also be mentioned.)
Minor points:
The sentence in lines 183 and 184 is repeated and should be deleted
Author Response
Reviewer 3
The paper by Possart et al. describes the identification of molecules inhibiting enzymes of Trypanosoma brucei and Leishmania major pteridine metabolism, specifically pteridine reductase (PTR1) and dihydrofolate reductase (DHFR). These enzymes are essential to the parasites and represent interesting therapeutic targets to treat sleeping sickness, cutaneous leishmaniasis, and other diseases caused by these trypanosomatid parasites. This is particularly important since the few currently available drugs exhibit severe side effects.
The study expands a previous work of a research group with extensive experience in in silico screening and in vitro selection of inhibitors. I enjoyed reading the manuscript. It is clear, well-written, elaborates well on the obtained data, and discusses them. The conclusions are well supported by the data, and the control experiments are appropriate.
The results are interesting, and the information provided can be useful for numerous researchers, as diseases caused by trypanosomatid parasites remain a serious health problem, and new, more specific drugs are urgently needed. Therefore, I recommend the publication of the manuscript after revision.
Points to improve before publication:
I would suggest including a more detailed description of the experiments needed to determine the type of inhibition of the identified products in the discussion section. While analyzing the effect of dilution of the enzyme preparation on the activity is useful, I would recommend describing other well-known methods (for instance, determining the kinetic parameters in the presence and absence of inhibitors. Vmax would decrease in the presence of an irreversible inhibitor, but Kms would remain unchanged. In this way, irreversible inhibitors behave as reversible non-competitive inhibitors. Other methods like gel filtration, dialysis, etc., could also be mentioned.)
We do agree with the reviewer that there are other possible experiments to elucidate the mechanism of action of irreversible inhibitors. We have duly mentioned in the text (several instances) that further investigations in this direction are in progress. We have extended the sentence in the discussion (page 8, end of first paragraph) by mentioning that the further studies are “including investigations on the inhibition kinetics, mass spectral measurements and site-directed mutagenesis experiments.” We feel that a much more detailed description at this point would not be useful before being able to actually report on the results. We have therefore decided to confine to this rather short description and to leave all further mechanistic work for a future publication dedicated specifically to this question. We hope the reviewer will be satisfied with this.
Minor points:
The sentence in lines 183 and 184 is repeated and should be deleted
The second sentence was deleted.
We thank the reviewer for the time and effort to help us improve our manuscript.
